# Cochaperones convey the energy of ATP hydrolysis for directional action of Hsp90

Leonie Vollmar [1,2,4], Julia Schimpf[1,2,4], Bianca Hermann [1] &
Thorsten Hugel [1,3] ✉

The molecular chaperone and heat shock protein Hsp90 is part of many protein complexes in eukaryotic cells. Together with its cochaperones, Hsp90 is responsible for the maturation of hundreds of clients. Although having been investigated for decades, it still is largely unknown which components are necessary for a functional complex and how the energy of ATP hydrolysis is used to enable cyclic operation. Here we use single-molecule FRET to show how cochaperones introduce directionality into Hsp90's conformational changes during its interaction with the client kinase Ste11. Three cochaperones are needed to couple ATP turnover to these conformational changes. All three are therefore essential for a functional cyclic operation, which requires coupling to an energy source. Finally, our findings show how the formation of subcomplexes in equilibrium followed by a directed selection of the functional complex can be the most energy efficient pathway for kinase maturation.

Being one of the most abundant proteins in the cytoplasm[1], the molecular chaperone and heat shock protein Hsp90 collaborates with more than 20 cochaperones to help other proteins (clients) acquire their active conformation[2,3]. The homodimeric Hsp90 is structurally well-equipped to assist numerous clients. Each Hsp90-monomer consists of three domains: The N-terminal domain (NTD) contains the ATP-binding site, the middle domain (MD), which is connected to the NTD by a charged linker sequence, is important for ATP-hydrolysis[4], and the C-terminal domain (CTD), which is the dimerization interface of highest affinity[4], contains a C-terminal MEEVD motif[5]. Due to its structural flexibility, the Hsp90-dimer can undergo large conformational changes ranging from an open, V-shaped structure to a tightly closed state[6–8], where also the middle and N-terminal domains dimerize. Additionally, Hsp90 is a member of the GHKL superfamily (Gyrase, Hsp90, Histidine Kinase, MutL) whose members share structural similarities, especially in the ATP-binding domain[9].

During the last years, several models were suggested in which Hsp90's client processing as well as related structural changes are linked to an ATP hydrolysis cycle. However, it is still unclear which components of the Hsp90 machinery are necessary for the efficient use of the energy from ATP hydrolysis. In fact, one of the long-standing

questions in the field is how ATP hydrolysis is utilized in the Hsp90 machinery at all. Similar questions have been answered for motor proteins[10], GTPases[11] or DNA gyrase[12], but not for chaperones, where a clear directional coordinate is missing. Even for specific complexes like the Hsp90-cochaperone-kinase machinery, the whole dynamic picture remains enigmatic and its depiction highly suggestive. Several publications propose, for example, that the complex closes upon ATP-binding[13] and then rearranges to a second, more compact closed structure[14]. Upon ATP-hydrolysis, the complex opens again and releases the maturated kinase as well as the cochaperone, leaving Hsp90 ready for yet another cycle[13,15–17]. Alternatively, it was shown that Hsp90 does not close upon ATP binding itself[18], but instead, Reidy et al.[19] proposed recently that its conformational changes are triggered by the proper positioning of ATP's gamma phosphate. Furthermore, it was proposed that the cycle might also be linked to dephosphorylation of a kinase[20]. Such cycles are a reasonable assumption but have not yet been shown experimentally. They imply directionality, i.e. energy consumption at constant turnover, which can only be proven with single-molecule experiments as detailed in the following.

To better understand the concept of directionality, it is helpful to introduce the terms detailed balance and steady state. For systems

[1]Institute of Physical Chemistry, University of Freiburg, Freiburg, Germany. [2]Spemann Graduate School of Biology and Medicine (SGBM), University of Freiburg, Freiburg, Germany. [3]Signalling Research Centers BIOSS and CIBSS, University of Freiburg, Freiburg, Germany. [4]These authors contributed equally: Leonie Vollmar, Julia Schimpf. ✉e-mail: th@pc.uni-freiburg.de

with at least two states, detailed balance can be defined, meaning that the populations $\pi$ of all states do not change over time. This even holds true if the forward and backward rates between the states are not equal. This seemingly counterintuitive statement is best explained in terms of different probabilities $p$. When looking at an example (Fig. 1) with a sparsely populated state 0 and a high transition probability (i.e. rate constant) to state 1 on the one hand, and a highly populated state 1 with a low transition probability back on the other hand, transition probability multiplied by the state population can result in the same 'amount' that transitions from state 0 to 1 as vice versa in a given time. This is a necessary condition for a system in detailed balance, namely that the forward 'amount' equals the backward 'amount' (and not the rate constant) along every edge ($i \leftrightarrow j$).

$$\pi_i p_{ij} = \pi_j p_{ji} \tag{1}$$

If that holds true for all edges, the populations of all states involved remain constant and are unchanged over time (stationary).

If there are more than two states, another possibility for stationary (constant) state populations is possible: steady state. In this case, e.g. more transitions occur from state 0 to state 1 than vice versa in a certain time. Those unequal forwards and backwards amounts are countered by transitions to other states of the cyclic system, allowing the population sizes of the states to stay the same. In our example (Fig. 1), we observe the binding and unbinding of three proteins $X$, $Y$, and $Z$. Initially, all proteins are unbound (state 0). When proteins $X$ and $Y$ bind, their subassembly corresponds to state 1. The resulting final complex $XYZ$ of all three proteins corresponds to state 2. It eventually falls apart again, causing the system to return to state 0. If the system was in detailed balance (Fig. 1a), equal amounts of complexes would form and disassemble in each step. The system would be in

thermodynamic equilibrium. However, with an energy source like ATP present, a succession of states could occur. In this scenario (Fig. 1b), the complex formation would be directed and a net flux through the cycle could be observed, with all forward amounts being larger than their backwards amounts. The state populations would still remain stationary (constant) but the system would be in a steady state. In a cellular environment, all states are populated, and a directed cycle allows for the recycling of proteins. Steady-state and detailed balance can only be clearly distinguished with single-molecule experiments because, from an ensemble view, all state populations stay constant in both cases. Another possibility to introduce directionality would be the successive binding of proteins by utilizing the energy of binding. However, this can never lead to a cycle as for every forward transition, a new protein would have to be synthesized to be pristine for binding. In the end, the very stable complex would have to be degraded because it constitutes the energy minimum of the binding process. This single-use of proteins would result in a huge energy cost and is therefore very unlikely.

Mathematically, directionality can be expressed in terms of Gibb's free energy $\Delta G_\circlearrowleft$ during one cycle, being different from 0.

$$\Delta G_\circlearrowleft = - \sum \ln\left(\frac{k_{ij}}{k_{ji}}\right) k_B T \tag{2}$$

In this equation, $k_{ij}$ is the forward rate from state $i$ to state $j$; $k_{ji}$ is the corresponding backward rate, $k_B$ is the Boltzmann constant and $T$ is the temperature. As another measure for directionality, the entropy production and flux can be calculated from the rates and the state populations in a steady state (see Supplementary Note 2 for details).

As mentioned above, directional cycles have often been suggested for the Hsp90 machinery, linking its conformational states to its ATPase function. Experimentally, several conformational states have

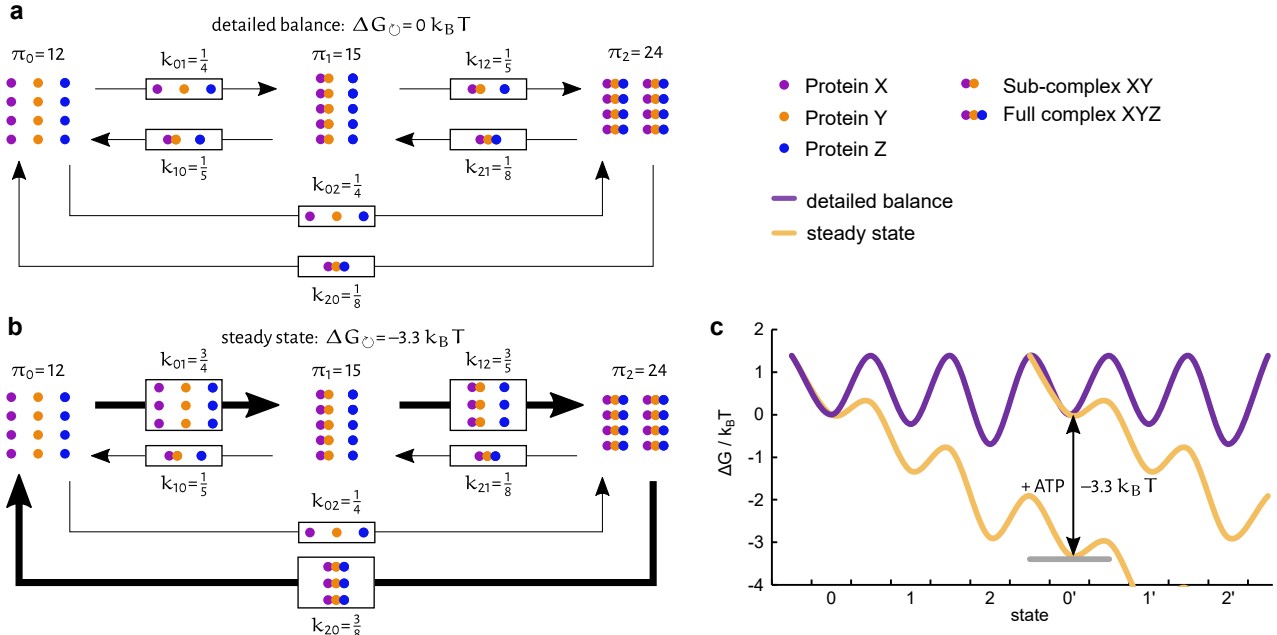

**Fig. 1 | Detailed balance and steady state.** Schematic representation of an exemplary system consisting of the three proteins $X$ (violet), $Y$ (orange) and $Z$ (blue) in detailed balance (**a**) or steady state (**b**). Grouped protein dots represent (sub-)complexes, $\pi_i$ signifies the state population of state $i$, the number of proteins equals the population size, and $k_{ij}$ denotes the kinetic rates between states $i$ and $j$. In detailed balance, the same number of proteins (here: 3) change their state at each node in every time interval, although the backward and forward rates are not equal (because the state populations are not equal). In a steady state, there is a net flux in

one direction (here: 9 forward, 3 backward, i.e. net flux of 6), which is represented by arrow width, but the state populations stay constant, as they do in detailed balance. To obtain this directionality, energy is needed. This is depicted in the energy landscape (**c**). Here, the purple line signifies the system from (**a**) in detailed balance, while the yellow line represents the steady state condition from (**b**). In the latter case, energy (e.g. in the form of ATP hydrolysis) is needed for the system to return to the initial state and start another directed cycle. In our examples, this energy is $-3.3 k_B T$.

been observed by x-ray crystallography and cryo-EM[6,21,22]. The dynamics between these conformational states can be measured with single-molecule Förster Resonance Energy Transfer (smFRET), similar to many other protein systems[23–26]. From the FRET efficiency, an open and closed conformation can be distinguished. Although its opening and closing dynamics suggest a two-state system, Hsp90 actually occupies at least four states, displaying two degenerate (with respect to FRET efficiency) open states (0 and 1) and two degenerate closed states (2 and 3) which differ kinetically. A separation of the degenerate pairs, however, requires the analysis of their kinetics. Previously, we have shown that these four states are hardly coupled to ATP hydrolysis for the idle Hsp90, i.e. in the absence of any other proteins[8,27].

Even though ATPase activity seems to be a prerequisite for client maturation, yeast Hsp90 displays a slow ATPase rate, only hydrolysing about one ATP per minute[28]. Here, we have added several co-chaperones (Aha1, Sba1, Cdc37) and the client kinase Ste11 to understand when the energy of ATP hydrolysis is coupled to Hsp90's conformational changes and therefore, when directionality is introduced. All of the cochaperones used within this paper are known to modulate Hsp90's ATPase activity: while Aha1 enhances it[29,30], Sba1[31–33] and Cdc37 have been shown to inhibit it[34]. In this context, Sba1 has been found to stabilize client-activating states, preferably binding to ATP-bound Hsp90[6,35]. Cdc37, on the other hand, has been dubbed a 'kinase-linking cochaperone' due to its specific and dynamic interactions with inactive kinases and Hsp90[15,16,21,36]. In this context, it is important to note that to function properly, Cdc37 needs to be phosphorylated, with Ser-14 and/or Ser-17 most likely being the relevant phosphorylation sites. It was also found that in vitro, the quasi-phosphorylated mutant Cdc37$_{S14,17E}$ shows activity[37].

Here, we show that the presence of Cdc37 with such phospho-mimics, the kinase Ste11 and ATP lead to changes in Hsp90's kinetic rates, but they do not introduce directionality. However, once Aha1 and Sba1 are added, clear directionality can be observed. Therefore, all three cochaperones are necessary to convey the energy of ATP hydrolysis.

## Results

### Directionality can be detected and quantified with single-molecule FRET experiments

To observe and quantify a system's directional behaviour on a single-molecule level, data acquisition, analysis and thereupon modelling require high levels of reliability—especially when the expected changes in Gibb's free energy are small. In fact, this has only been done for proteins that move either linearly like kinesin or myosin[38,39], rotate like the F$_0$F$_1$-ATPase[40], or change the length of a substrate like DNA gyrase[12].

To address the challenge of observing directionality in conformational changes, we have developed a testing procedure that uses external laser triggering to create artificial directionality in experimental data. Therefore, we created an artificial smFRET system using labelled low-FRET dsDNA[41]. With a total internal reflection fluorescence (TIRF) microscope for data acquisition, dwell time analysis and Hidden Markov Modelling (HMM) for analysis and evaluation, we first checked whether directionality could be retrieved from single-molecule experiments. By laser triggering, we created a loop of four different states with a defined sequence (Fig. 2a): a long-lived high-FRET state followed by a long-lived low-FRET state, then a short-lived high-FRET state before a short-lived low-FRET state. While the lifetime of the states was controlled by illumination time, the difference in FRET efficiency was achieved by using an additional laser. When our low-FRET dsDNA sample is simultaneously excited by the donor (green) and the acceptor (red) laser, it artificially displays high FRET by direct excitation. In contrast, the dsDNA shows low FRET if only excited with the green laser. A dwell time analysis was performed on the single-molecule signal. The two apparent states (high vs. low) were

allocated from the signal intensity (Fig. 2b). The two different lifetimes of 1.5 and 5 s within these states can be inferred from the dwell time histograms (Fig. 2c, d), which allowed us to deduce four different states. By following the pattern of the trace and considering the different lifetimes, the sequence of those states can be inferred. This sequence matches the ground truth introduced by laser triggering well.

In addition, we did an HMM analysis via SMACKS[27] on this artificial smFRET data. Although the prerequisite of stochastic switching for Markov Modelling is not given here, the analysis worked very well (see Supplementary Note 1 for a more detailed discussion and results). The deviations of the populations are <4%, while the transition rates deviate by a maximum of 8%. Although only a forward progression through the cycle was triggered, the analysis retrieved a few false backward rates due to wrong state allocation. The resulting very small backward rates allowed for an $\Delta G$ calculation amounting to −165 $k_BT$. In contrast, the calculation of a $\Delta G$ for the ground truth is mathematically not defined as there are no backward rates. Altogether, these results show that directionality can be detected and analysed on the single-molecule level.

To quantify the directionality that can be retrieved by our software SMACKS and to compare it to another software (Hidden-Markury[42]), we used simulated data with a directionality of 0 $k_BT$, −3 $k_BT$ and −10 $k_BT$, respectively (Fig. 2e, f and Supplementary Fig. 1). Additionally, two data sets with the same $\Delta G$ value of −2 $k_BT$ but with different transition rates (one with rates similar to rates measured for Hsp90, and the other with equally distributed state populations) were simulated (Fig. 2e, f). The data was simulated with MASH-FRET[43] and then analysed with SMACKS and Hidden-Markury. Both programmes, which have previously been tested in a comparative study[44], were able to retrieve the pre-specified $\Delta G$s. For $\Delta G = 0$ $k_BT$, Hidden Markury obtained an $\Delta G$ of 0.505 $k_BT$, while SMACKS resulted in an $\Delta G$ of 0.02 $k_BT$. This important control confirmed that for data representing a system in detailed balance, none of the programmes overestimated the $\Delta G$ values of the ground truth, i.e. resulted in a falsely positive directionality. For $\Delta G = −2$ $k_BT$ (Hsp90 like), Hidden Markury recovered 93% of the ground truth while SMACKS recovered about 80%. In all cases with a $\Delta G \neq 0$, the directionality was underestimated by a maximum of 25%. Some underestimation is expected, as the result from Hidden Markov Modelling is only a lower limit[45]. Altogether, our findings show that we can distinguish systems in detailed balance from those with a directional flux—even if the analysed system does not show strong directionality, i.e. small |$\Delta G$|s.

### Directionality in the Hsp90-kinase machinery requires three cochaperones

After having ensured the robustness of both data acquisition and analysis, our experiments were extended to the Hsp90 machinery. Figure 3a, c shows examples of dynamic smFRET traces of Hsp90's opening and closing dynamics in the presence and absence of cochaperones and a client kinase (more traces in Supplementary Fig. 2). The FRET pair positions chosen on Hsp90 (61 and 385, one on each monomer) are well-established for these experiments since they show a substantial difference in FRET efficiency between Hsp90's closed and open conformation[8,27,46]. Hidden Markov Modelling (HMM) shows that Hsp90's dynamics are best described by a four-state model (see Supplementary Fig. 3), having two open and two closed states. Both open as well as closed states are degenerate in their FRET efficiency but can be separated by their kinetics. Here we refer to Hsp90's open states as states 0 and 1, and its closed states as states 2 and 3. In general, forward and backward rates for transitions between all of them can be obtained by kinetic analysis with SMACKS[27]. However, diagonal rates (02, 20, 13 and 31) are always close to zero and are therefore not considered in the following (see Supplementary Fig. 3). In previous experiments without an energy source present, i.e. conditions where no directionality is

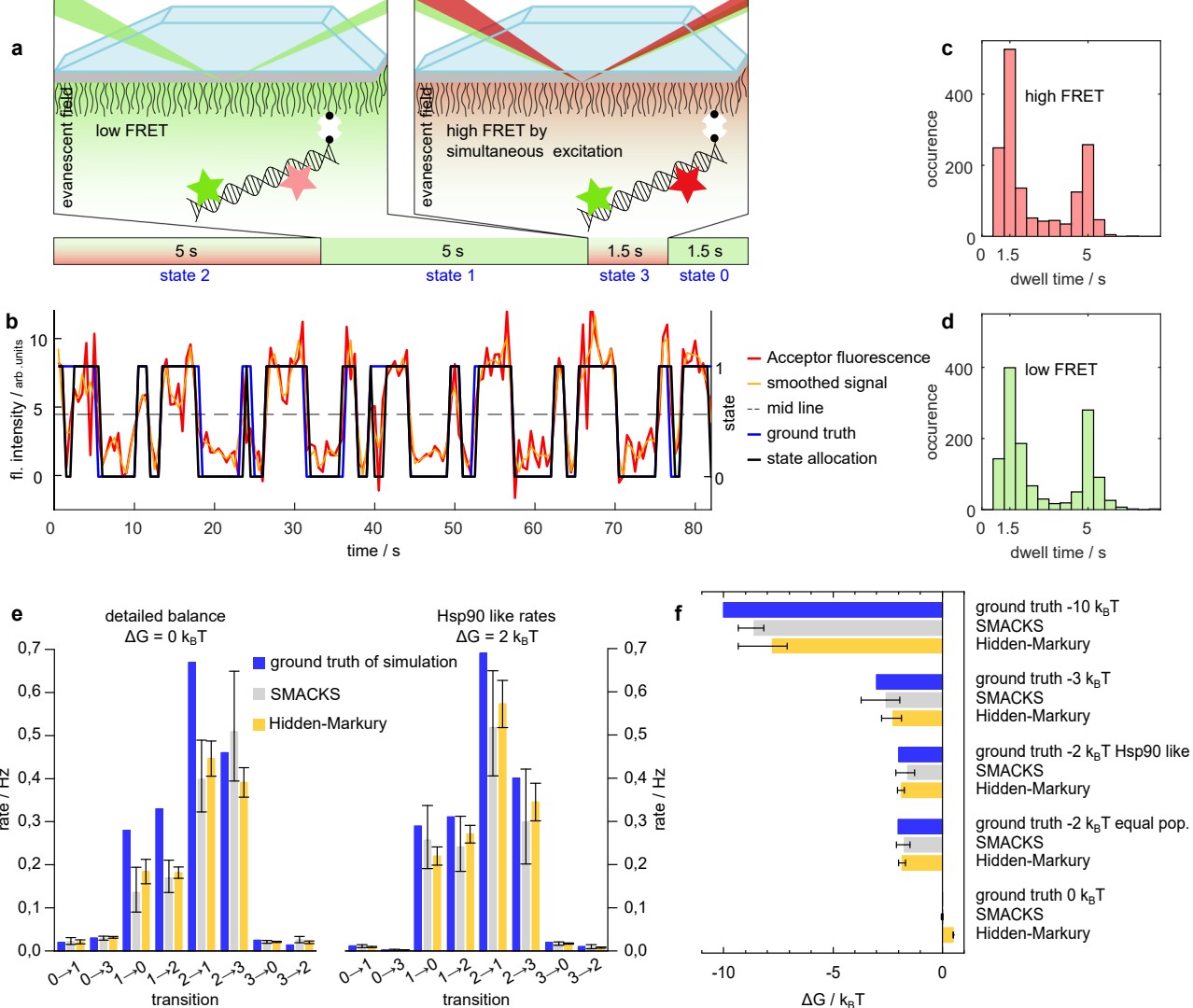

**Fig. 2 | Our single-molecule detection and analysis is capable of quantifying directionality. a** Artificially introduced directionality by laser triggering. To achieve two distinct FRET efficiencies, a fixed low-FRET dsDNA sample is illuminated alternately with a green laser only (low FRET) or with a red and a green laser simultaneously (artificial high FRET). Lifetime of the states can be controlled through different illumination times (1.5 vs. 5 s). **b** Single-molecule trace (red) of acceptor dye after donor or simultaneous excitation, respectively. States were allocated (black) according to their position respective to the midline (black dashed) of the signal smoothed with a Savitzky–Golay filter (window = five frames). Pattern of the laser triggering in blue. **c** and **d** Dwell time histograms from 349 single-molecule traces according to their allocated state. **e** Rates for simulated traces (200 each) of a four-state model in detailed balance or with a directionality of $-2k_BT$. Ground truth in blue, the rate constants are very similar to Hsp90's behaviour. Results from HMM analysis with SMACKS (grey) and Hidden-Markury (yellow). Error bars show 95% CIs. **f** Retrieved Gibb's free energy from the analysis of five simulated conditions (ground truth, blue) with SMACKS (grey) and Hidden-Markury (yellow) Error bars show 95% CIs. All values are given in Supplementary Table 3 and in source data which is provided as a Source Data file.

expected, we measured $\Delta G$ values ranging from 0.3 to −0.9 $k_BT$ (see Supplementary Table 2). Therefore, these measurements serve as the lower limits of what can be considered directional, i.e. we consider any experiment with a $|\Delta G|$ above 1 $k_BT$ as clearly directional. Please note that the type of sign (positive or negative) only indicates the direction of the cyclic movement (clockwise or counter-clockwise).

First, we tested if Cdc37, Ste11 and nucleotide introduce directionality, which is the currently prevailing model (conditions with one Cdc37 per Hsp90 dimer (Cdc37$_1$-Hsp90$_2$) were used). Figure 3a shows an exemplary smFRET trace and Fig. 3b the extracted model. Note that the Viterbi path (black trace) is not a fit of the data, but the model extracted from the data, with the 'state' axis on the right. Figure 3e shows clear changes compared to idle Hsp90 in the transition rates 10, 01 and 12. Rate 21 is increased as well but within the 95% confidence interval. However, despite noticeable rate changes, the system stays in detailed balance, i.e. there is no directionality (Fig. 3e and g).

Therefore, we tested if the addition of further cochaperones− Aha1 and Sba1−to the system introduces a directed progression of Hsp90 through its conformational states (Fig. 3c and d). Indeed, in this case not only the transition rates are changed, but the system also shows Gibb's free energy of $\Delta G = -2.1\ k_BT$, meaning that the system is no longer in detailed balance. This behaviour is mostly due to a strong increase in rate 23 (Fig. 3e and g).

As such calculated free energy is a lower bound for the energy coupled into the open-close movement, this constitutes evidence that the energy of ATP hydrolysis couples to the large conformational changes of Hsp90, although only in the presence of three cochaperones and a substrate. In further experiments, the effect of either Aha1 or Sba1 on the minimal model was tested as well, however, no directionality was induced by the sole addition of one interaction partner (Fig. 3g and Supplementary Fig. 4). This indicates that the Hsp90 machinery relies on a larger number of interaction partners to escape

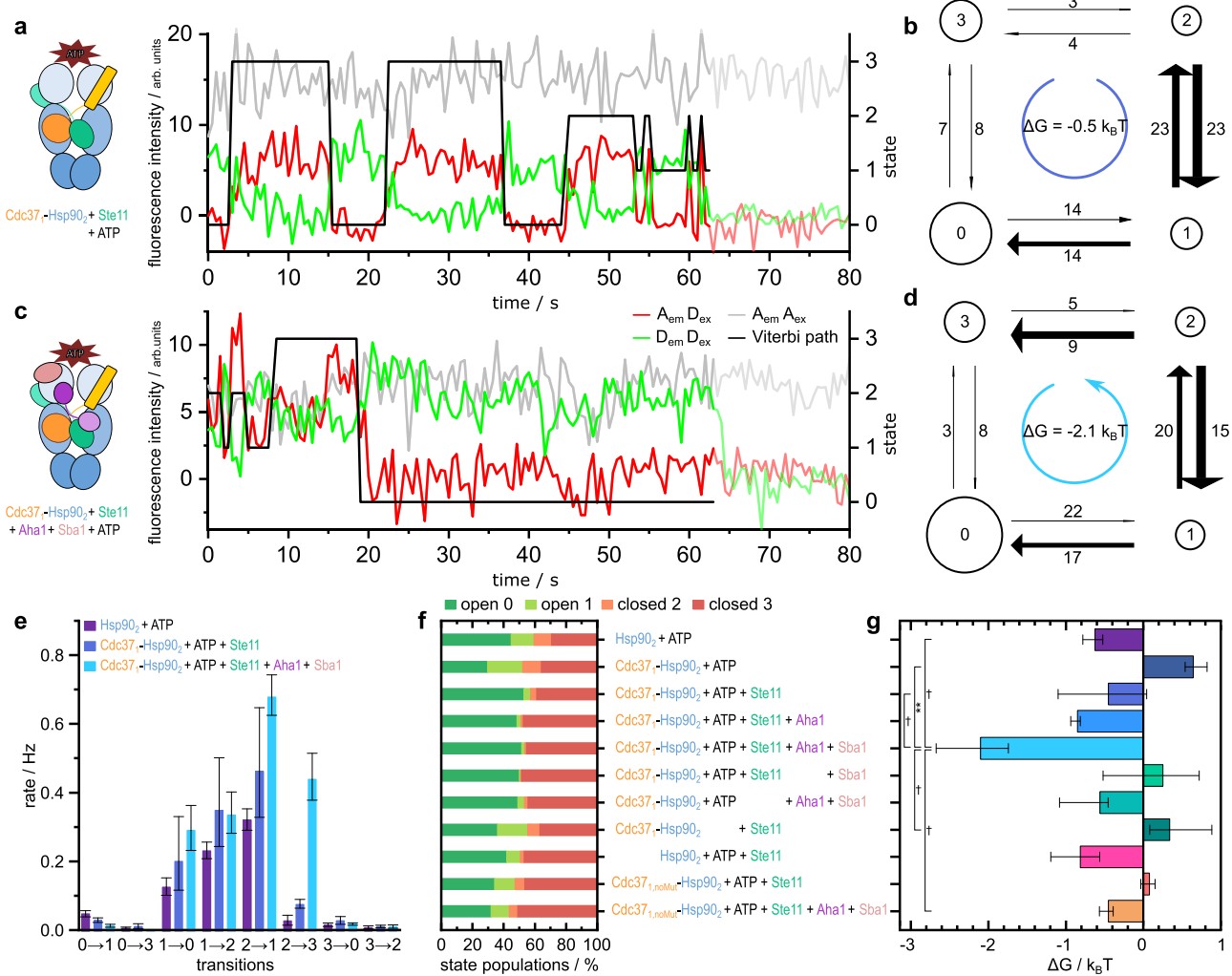

**Fig. 3 | Three cochaperones are needed to induce directionality.** Hidden Markov analysis of single-molecule traces (**a, c**) determines Viterbi paths (black line), transition rates (**e**) and state populations (**f**), which can be represented as models (**b, d**) and allow for calculation of ∆G (**g**). **a, b** Cdc37$_1$-Hsp90$_2$ + ATP + Ste11 in detailed balance: a directed progression through Hsp90's conformational states can neither be observed in the Viterbi path (**a**) nor the corresponding model (**b**) (n = 509 traces). **c, d** Cdc37$_1$-Hsp90$_2$ + ATP + Ste11 + Aha1 + Sba1 in steady state: single-molecule trace displays a directed Viterbi path through all states, starting with fluctuations between state 1 and 2, then transitioning to long closed (state 3) and afterward to long open (state 0) (n = 998 traces). **b, d** Four-state models with equal FRET efficiencies for states 0/1 (open) and 2/3 (closed), respectively. Circle sizes are proportional to state population and arrow widths to transition rates. Numbers on arrows indicate the normalized amount of transitions occurring.

**e** Overview of transition rates ± 95% CIs of selected conditions, all rates are given in Supplementary Fig. 4 (n$_{Hsp90+ATP}$ = 874 traces). The presence of all three cochaperones (lightest blue) leads to a strong increase in rate 2 → 3. **f** Percentage of state populations for each measured condition. The presence of clients and cochaperones causes longer dwell times in the long-lived states 0 and 3. **g** Overview of calculated Gibb's free energy for each measured condition. Directionality can only be observed in the presence of all three cochaperones (lightest blue). Two-sided Student t-tests with Holm–Šídák correction for multiple comparisons were performed, selected p-values shown **p < 0.01, †p < 0.1. Results shown in **e–g** are obtained from two to three independent repetitions with 321–998 traces in total. Exact numbers for each condition and corresponding confidence intervals for rates and for ∆Gs are given in Supplementary Table 1. Source data are provided as a Source Data file.

detailed balance and enter steady state and therefore to convey the energy of ATP hydrolysis.

Additionally, the effect of Cdc37 without the phosphomimicking mutations S14,17E (Cdc37$_{noMut}$) on Hsp90's conformational behaviour was tested. Neither for the condition formerly showing directionality (i.e., in the presence of Ste11, ATP, Aha1 and Sba1) nor for the minimal model (i.e., in the presence of Ste11 and ATP), directionality could be retracted. This is in line with previous research[37] stating that (quasi-)phosphorylation of Cdc37 is necessary for its activity (Fig. 3g).

In contrast to the detectable changes in its kinetic rates, a look at Hsp90's state populations reveals that none of the examined conditions significantly affected thermodynamics: Hsp90's different interactors do not initialize a closing of the dimer, i.e. the ratio of open to closed Hsp90 was not strongly altered (Fig. 3f). This is especially

striking for the presence of Cdc37, as it inhibits Hsp90's ATPase rate (Supplementary Fig. 5). This again indicates that Hsp90's conformational state is hardly connected to ATP hydrolysis. Even more astounding: Aha1's induced closing of the dimer[47] is abolished in the presence of Cdc37 and Ste11. The presence of Cdc37, Ste11 and ATP (both in the presence and absence of Sba1 and Aha1), 'locks' almost half the complexes in the longer-lived states 0 and 3, respectively. This can be explained by the strongly increased rates 10 and 23. The enhancement of the opening rate 21 adds to this as well but is compensated for by the simultaneous rise of its opposed closing rate 12, meaning that the exchange between the short-lived states is accelerated.

Dimer closing can also be triggered by non-hydrolysable ATP analogues like ATPγS[48]. This is also the case in the presence of Cdc37, Ste11, Aha1 and Sba1 (Supplementary Fig. 6a). As ATPγS keeps Hsp90 in

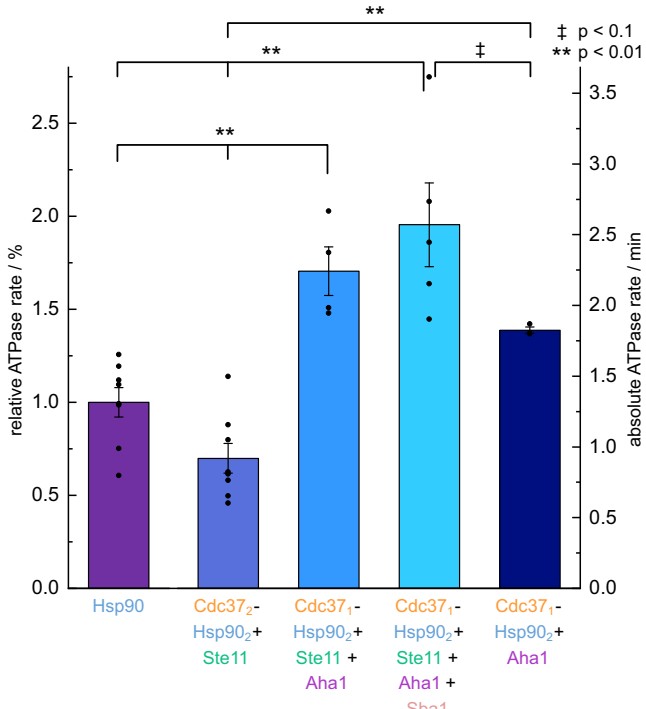

**Fig. 4 | ATPase rates of Hsp90.** Relative and absolute ATPase rates ± SEM of Hsp90 in the presence and absence of the cochaperones Cdc37, Aha1 and Sba1 and the client kinase Ste11. Hsp90 is a slow ATPase with 1.3 hydrolysed ATP per minute (violet). Ste11 hinders Cdc37's ATPase-decreasing effect on Hsp90 (blue, second to left bar). Aha1, Cdc37, Sba1 and Ste11 together (light blue, second to right bar) strongly increase the ATPase to 2.6 ATP/min. This effect is not achieved by Cdc37 and Aha1 alone (darkest blue). Statistical significance was tested by one-way ANOVA and Tukey post hoc test (described in methods, ($F(8,58) = [23.41464]$, $p = 0.05$)). Additional conditions are shown in Supplementary Fig. 5. Source data are provided as a Source Data file.

the closed state, the movement through a directional cycle is consequently abolished (Supplementary Fig. 6b, c).

Altogether, the changes in Hsp90's kinetic rates indicate a cooperative interplay between Cdc37 and Ste11 in the presence of ATP. These changes, however, are not yet sufficient to introduce directionality. For an ATP-driven cycle of Hsp90's conformational kinetics—and therefore the cyclic operation of Hsp90—additional Aha1 and Sba1 are required, suggesting their active role in conveying the energy provided by ATP. Please note that the alternative to a cyclic operation is an energetic downhill process through successive binding of proteins. This is very unlikely because it would mean that one Hsp90 has to be degraded at the end of the energetic downhill process for each kinase activation event.

**Directionality and ATPase rates are only weakly linked in Hsp90-kinase complexes**

In the following, we tested how the directed progression of Hsp90's dynamic, i.e. cycling, is linked to Hsp90's ATPase rate. Therefore, we determined Hsp90's ATPase in the presence of Cdc37, Sba1, Aha1, Ste11 and their combinations. Hsp90's ATPase rate was found to be 1.3 ± 0.3 ATP/min, fitting well to previous data[28]. Figure 4 shows that in the presence of equimolar amounts of only Cdc37 and Ste11—a condition that did not induce directed behaviour of Hsp90's kinetics—the chaperone's ATPase rate is not significantly altered. This implies that there is an interaction between the kinase and Cdc37, preventing the latter from performing its full inhibitory function (Supplementary Fig. 5). For human Hsp90, it was previously shown that the sole addition of a client (the ligand-binding domain of the glucocorticoid receptor) is sufficient

to control Hsp90 ATPase activity[49]. However, the presence of equimolar amounts of Ste11 did not increase the ATPase of the Hsp90 yeast homologue used within this paper (Supplementary Fig. 5). In contrast, the presence of Aha1 enhances this rate in all of the following cases, albeit to a different degree: when Cdc37 is present as well, the rate is increased to 1.8 ± 0.2 ATP/min. When Ste11 is added to the mix, it is further enhanced to 2.2 ± 0.2 ATP/min. Surprisingly, upon the addition of Sba1—a known ATPase inhibitor—Hsp90's ATPase rate is boosted further to 2.6 ± 0.3 ATP/min. This final observation is in good agreement with the directionality observed in single-molecule measurements, indicating that there is some, but no strict, interplay between the directionality and Hsp90's ATP hydrolysis rate. Altogether, such an interaction between three cochaperones allows for more evolved regulation, e.g. through a typical nucleotide exchange factor.

## Discussion

Most suggested conformational cycles of Hsp90 imply strong directionality as they are commonly depicted as being unidirectional. Here we show a fundamentally different scenario, in which an upstream (i.e. preceding) equilibrium allows for an assembly of most of the involved proteins, and then one (or few) directed steps stabilize the functional complex (Fig. 5). We consider this scenario to be much more likely for the formation of any multi-protein complex for several reasons. First, for an energetic reason: Consider a complex of five molecules with a defined binding order. To achieve a reasonably defined binding sequence, the energy equivalent of at least one ATP hydrolysis (around 20 $k_B T$) needs to be consumed in each step, i.e. the hydrolysis of five ATP in total. Another possibility would be to provide this energy in the form of free energy of binding. However, this would eventually result in a very stable multiprotein complex that could only be broken apart again upon an energy input corresponding to five ATP—one for every step—or by degradation of some of the proteins involved. In our opinion, both options seem unlikely, as they would be highly inefficient and an Hsp90 dimer binds (and hydrolyses) two ATP at a maximum per cycle. In contrast, an upstream equilibrium preceding one or a few directed steps, as suggested here, would only involve energies of a few $k_B T$s. The final formation of the functional complex would then involve only stabilizing energy of e.g. one ATP hydrolysis. Therefore, the components could be recycled by the hydrolysis of one or two ATP, which are available from Hsp90.

A second reason is the timescale of complex formation: If this were a strongly directed downhill process, it could easily happen in a fraction of a second (motors like myosin, kinesin or the $F_0F_1$-ATPase hydrolyse ATP with up to several hundred per second). This, however, is not necessary in our opinion. Dynamic complex formation on the timescale of seconds allows for much more regulation, e.g. by post-translational modifications (PTMs). Third, our data clearly shows that there is no linear succession of binding events. A linear succession of binding events (i.e. breaking of detailed balance) is only possible in two ways: a) In a cyclic manner, with the energy source being coupled to the cycle, which was not the case here as it would need more energy than the observed −2.1 $k_B T$, and b) by the production and degradation of a protein for every step in the linear succession. This would be an extreme waste of energy and therefore considered highly unlikely. To our knowledge, there is also no published study that shows such a succession. Of course, if components are added in a certain order, an artificial succession is introduced, however, this does not serve as evidence for the succession of binding in steady state or even living cell conditions. Actually, such an artificial introduction of proteins would correspond to protein production and degradation in every cycle.

Up to this point, we have neglected conformational changes, which define multi-conformational complexes[50]. It is important to note that not only Hsp90 samples different conformations, but also its cochaperones. Their thorough study is therefore necessary for a

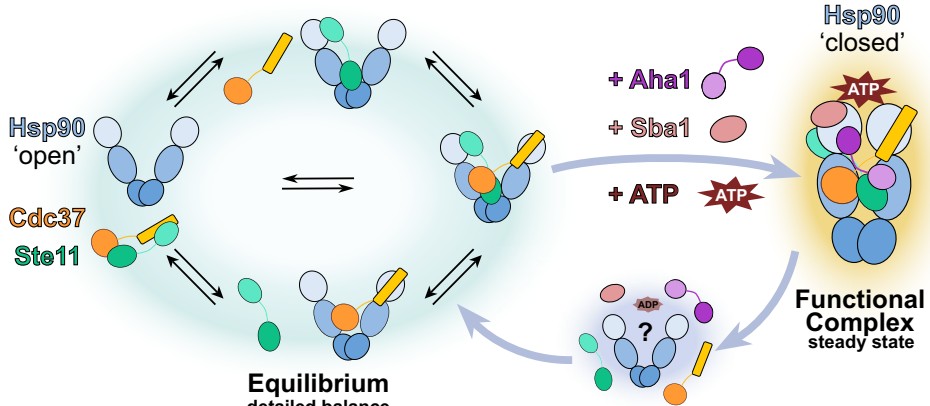

**Fig. 5 | Assembly mechanism and the conformational cycle of Hsp90.** From sub-complex assemblies in equilibrium (green halo) to the functional multiprotein-Hsp90 complex (orange halo). Sub-complexes of Hsp90, Cdc37 and Ste11 are in detailed balance, thus assemble and disassemble in equilibrium. Upon the addition of Aha1, Sba1 and ATP, the functional complex is stabilized, e.g. by utilizing the energy provided by ATP binding. Therefore, recycling of the components and complex disassembly (blue halo) would only require the energy provided by hydrolysis of one ATP. The third state, where all proteins are unbound (blue halo), was not quantified and could also be part of the left equilibrium, but more than two states are necessary for directionality (blue arrows). Another possibility for this third state could be a partially disassembled state with hydrolysed ATP (ADP).

complete understanding of the machinery. Cdc37, for example, was shown to change between a dynamic extended conformation and a compact structure to sense partially unfolded clients and then stabilizing them[16]. Aside from structural rearrangements, phosphorylation plays an important role in Cdc37's functionality as a part of the Hsp90-chaperone machinery. In line with previous findings[37], we showed that only the quasi-phosphorylated mutant $Cdc37_{S14,17E}$ would lead to directional behaviour (in the presence of other cochaperones, kinase and ATP). In this context, a directional cycle might also be obtained by looking at $yCdc37_{S14}$'s dephosphorylation by protein phosphatase 5 (PP5 in humans, Ppt1 in yeast), which is thought to be important for kinase release[51,52].

If these conformational changes and PTMs were also included in the successive, directed assembly, even more energy would be required. In addition, we and others have shown before that Hsp90's conformational changes occur on much faster timescales than ATP hydrolysis[8,27]. Thus, conformational changes are thermally driven and not fuelled by ATP turnover. A recent study suggests that it is not the hydrolysis of ATP but the repositioning of its gamma phosphate which is sufficient for Hsp90 function[19]. They emphasize that returning to the open, ADP-bound state is essential, which can occur either by nucleotide exchange (non-directional) or by hydrolysis (directional). ATP hydrolysis is evolutionarily preferred because it allows for a regulatory switch that can be tuned by cochaperones. By regulating the ATPase activity, cochaperones can regulate the dwell times of Hsp90's conformational states. In agreement with this observation, we see a shift of dwell times to the long-lived conformational states in our single-molecule experiments (Fig. 3f).

On a related note, it is worth comparing these findings to other members of the GHKL ATPases, especially DNA gyrase, which has a structurally highly similar ATP binding domain. Similarly to Hsp90's conformational behaviour, it was previously shown that gyrase's structural transitions are not strictly dependent on ATP hydrolysis but are rather loosely coupled[37]. When processing DNA, both conformational states of the gyrase prevail in the absence of the nucleotide. However, the presence of ATP accelerates the transition from one to the other and funnels the system towards a directional cycle through loosely coupled transitions[53]. This fits nicely with our findings and our interpretation (Fig. 5), where weakly coupled transitions (equilibrium, left) are funnelled towards a directional cycle by Cdc37, Aha1, Sba1 and ATP. Such a loose coupling between structural transitions and chemical sub-steps might therefore be a hallmark of GHKL ATPases in

general, as opposed to the tightly coupled and well-studied P-loop NTPases[54].

In summary, our data is in perfect agreement with an assembly mechanism as depicted in Fig. 5. No directionality can be detected in any sub-system of the five components Hsp90, Cdc37, Aha1, Sba1, Ste11, i.e. they are close to thermal equilibrium. Only when all five components are together and ATP is present, directionality can be detected. This would allow first for the formation of a functional complex and then, e.g. after ATP hydrolysis or nucleotide exchange[19], for recycling of all components. We speculate that this is the long searched-for function of Hsp90's ATPase function, namely to disassemble a functional complex after the job is done. Checkpoints might be included without the need for large amounts of free energy.

## Methods

### Expression constructs

All proteins used in this study originated from *Saccharomyces cerevisiae*.

Hsp90 constructs contained the *hsp82* gene from *Saccharomyces cerevisiae*, an N-terminal cleavable $His_6$-SUMO-tag and a C-terminal coiled-coil domain ensuring stable dimer formation[8]. The coiled-coiled motif is either followed by a Strep-tagII or an Avi-tag for in vivo biotinylation.

The basic Cdc37-Hsp90 fusion protein was purchased as a synthetic gene construct (GeneScript). Such fusions provide for a high local concentration and overcoming low binding affinities in single molecule experiments[55]. The linker between the two proteins consists of in total 115 amino acids, with the following sequence: P-$(GGS)_{15}$-P$(EAAAK)_3$-P-$(GGS)_{15}$-P[56]. The construct also carried the C-terminal coiled-coil motif followed by either a Strep-tagII or Avi-$His_6$-tag for in vivo biotinylation.

The Cdc37 domain from the fusion protein additionally carried two point mutations, S14E and S17E, the first glutamate mimicking a phosphorylated serine residue, the second was introduced due to the presence of natural glutamate forming a stabilizing salt bridge, as derived from a Cryo-EM complex structure[21]. In the free Cdc37, only the S14E phosphomimic was present.

All expression constructs were cloned into pET28b plasmid vectors and single cysteines in Hsp90 at position 61 or 385 were introduced in every construct for site-specific labelling. Free Aha1, Sba1 and Cdc37 constructs carried an N-terminal $His_6$-tag for purification.

## Protein preparation

Expression and purification were performed in variations of established protocols as previously reported[46,57]. In short, proteins were produced in *E.coli* BL21Star (DE3) or BL21 (DE3) cod+ and purified depending on the affinity tag. For in vivo biotinylation on the Avi-tag the plasmid pBirA (Avidity Nanomedicines, La Jolla, CA) was cotransformed and the cell culture procedure was adapted according to Avidity's in vivo biotinylation protocol.

His$_6$-SUMO tagged Hsp90 was isolated with affinity chromatography using a Ni-IMAC column (HisTrap HP), followed by SUMO cleavage with SenP protease and a second Ni-IMAC to separate the His-tag free protein from uncleaved proteins and the His$_6$-SUMO-tag. The protein was then applied to an anion exchange column (HiTrap Q, Cytiva) and finally to a size exclusion column (Superdex 200, Cytiva).

Strep-tagII and simple His$_6$-tag preparations were performed with a two-step protocol, starting with affinity chromatography using either a Strep-Tactin column (Strep-Tactin Superflow, IBA Lifesciences) or a Ni-IMAC (Cytiva/ Ni-NTA Agarose, Qiagen) followed by Size exclusion chromatography (Superdex 200, Cytiva).

## Protein Labelling and monomer exchange

Hsp90$_2$ and Cdc37$_1$-Hsp90$_2$-fusion were labelled with maleimide fluorescent dyes (ATTOTEC, Atto550 and Atto647N), as they allow for specific attachment to the introduced cysteines. First, the respective protein sample was incubated with 10 mM tris(2-carboxyethyl)phosphine (TCEP) in a volume of 50 μl and a protein concentration of 100 μM. After 20 min of incubation at room temperature, TCEP was removed by washing the sample five times with phosphate buffer (100 mM from 45 mM Na$_2$HPO$_4$ and 55 mM NaH$_2$PO$_4$, pH 6.7) in a centrifugal filter (Amicon Ultra, cutoff 30K). The sample was then incubated, using either Atto550 or Atto647N (1.5 times molar excess). After two hours at room temperature, the free dye was removed using a spin column (PD 25 MiniTrap, GE Healthcare) preequilibrated with measurement buffer (40 mM HEPES, 150 mM KCl, 10 mM MgCl$_2$, pH = 7.4). The labelling efficiency was determined using a Nano-Drop spectrometer (Thermo Scientific) and usually reached between 80% and 100%. As the dynamic rates between two point mutations of Hsp90 (D61C and Q385C) were monitored, monomer exchanges were performed by incubating a mixture of both variants for 40 min at 43 °C and 300 rpm in a shaking incubator (Thermomixer 5436, Eppendorf) which allows the opening of the coiled-coil domain and subsequent dimer formation upon cooling down to RT. This method was also employed to obtain Cdc37$_1$-Hsp90$_2$-fusion heterodimers (one Cdc37 per Hsp90 dimer). In the case of the single-molecule experiments, the mixtures contained a ratio of 1:2 of biotinylated to nonbiotinylated protein, as for these measurements only the biotinylated heterodimers would bind to the chamber surface. That way, the biotinylated protein has a higher chance of being exchanged with a nonbiotinylated variant. For the ATPase assays, the exchange ratio between both variants was 1:1 resulting in a binomial distribution of one part Cdc37$_2$-Hsp90$_2$, two parts Cdc37$_2$-Hsp90$_2$, and one part Hsp90$_2$.

## Single-molecule measurements

The measurements were conducted at a custom-built prism-type TIRF setup including two lasers of wavelength 532 nm (green, Coherent OBIS LS) and 637 nm (red, Coherent OBIS LX) as introduced in ref. [58]. The laser beam is focused perpendicularly onto a prism, which is located on top of the measurement chamber. To avoid unspecific protein binding to the flow chamber, its surface is passivated with an 80:3 mixture of methoxy-polyethylene glycol–silane (5000 Dalton, Rapp Polymere) and silane-PEG-Biotin (3000 Dalton, Rapp Polymere). To further enhance this passivation, the measurement is preceded by 30 min of BSA incubation (0.5 mg/mL in measurement buffer, Carl Roth). The biotinylated protein sample is attached to the surface via biotin-Neutravidin binding. Therefore, the chamber was further coated with Neutravidin (0.25 mg/mL, Thermo Fisher Scientific) previous to the addition of the sample (picomolar range). Free cochaperones (Aha1 and Sba1) and Ste11 (*Saccharomyces cervisiae* Ste11-(AA 415-712)-P23561-partial protein, Cusabio, washed with measurement buffer) were added at 2 μM each with 2 mM ATP present.

The sample was measured in alternating laser excitation (ALEX) mode between the green and the red laser, with excitation times of 200 ms and dark times of 50 ms. Fluorescence signal was collected by an oil immersion objective (×100 magnification, Nikon) and recorded by EMCCD cameras (iXon Ultra 897, Andor) at 3 × 3 binning. Image registration with fluorescent beads (TetraSpeck microspheres, 0.2 μm, Invitrogen) preceded the experiments to align the green and the red channel.

An Igor Pro (version 6.37) based in-house script was used for the selection of single-molecule traces. The programme identifies the positions of single-molecules by searching for the brightest spots in five consecutive frames of the respective detection channel. Single-molecule traces were corrected for leakage (0.16 ± 0.03), direct excitation (0.12 ± 0.02), $\gamma$ (1.09 ± 0.08), and $\beta$ (0.86 ± 0.11) as described before[41,59].

## Laser triggering for artificial directionality

Artificial directionality for two-colour single-molecule Förster resonance energy transfer (FRET) was induced by laser triggering and recorded in ALEX mode. The repeating loop consisting of a short-lived high-FRET state followed by a short-lived low-FRET state, then a long-lived high-FRET state before a long-lived low-FRET state was programmed in LabView 2019 (see Source Data). The length of both short-lived states was set to three consecutive frames, whereas for both long-lived states, a duration of 10 frames was chosen. The difference in FRET efficiency was achieved by using a labelled dsDNA sample displaying a low FRET efficiency ($E = 0.15 \pm 0.02$, 1-lo sample[41], biomers.net GmbH Ulm) and switching between excitation by the green laser only (low FRET) and simultaneous excitation by the green and the red laser (high FRET).

## Dwell time analysis of single-molecule traces from laser triggered experiment

The single-molecule intensity traces showing acceptor fluorescence after donor and simultaneous excitation, respectively, were smoothed by a Savitzky-Golay filter (window = five frames). States were allocated trace-wise according to their position respective to the mean signal intensity of the smoothed signal. Dwell times were counted; bin width of histogram is one frame (0.5 s).

## Simulation of dynamic time traces

Simulation of dynamic time traces was done with MASH-FRET[43]. For all simulations, four FRET states with two degenerate low as well as two degenerate high FRET efficiencies ($E_{low} = 0.1 \pm 0.05$, $E_{high} = 0.8 \pm 0.05$) were given. The transition matrices for Gibb's free energies of $-3k_BT$, $-10k_BT$, $0k_BT$ and two times $-2k_BT$ were predefined (Supplementary Table 3). 200 traces with 200 frames with a frame rate of 2 Hz were simulated, respectively.

## Single-molecule data analysis

The dynamic analysis of both simulated and experimentally measured data was done using two programmes employing Hidden Markov Models. In SMACKS[27], an Igor Pro (version 6.37) based script, two apparent states were chosen for FRET state assignment. A detailed description can be found in the Supplementary Methods. A four-state model with two hidden states was chosen by Bayesian information criterion (BIC) and subsequently a transition matrix was calculated (see Supplementary Fig. 3). In Hidden Markury, a python-based notebook[42], the same four-state model was assumed and diagonal rates were set to

zero. For rate calculation, the 2D model was used, which retrieves the input from donor and acceptor signal.

Mean transition rates were calculated for each condition from $n$ replicates (Supplementary Table 1). 95% confidence intervals (CI) resulting from each HMM modelling were treated as standard error of the means and averaged as such:

$$\overline{\text{length of CI}} = \frac{1}{\sqrt{n}}\sqrt{\frac{\sum_{i=1}^{n}(\text{length of CI}_i)^2}{n-1}} \qquad (3)$$

The Gibb's free energy $\Delta G$ for each protein condition was calculated from the averaged transition rates using Eq. (2). Error values were obtained by calculating $\Delta G$ of the averaged rates ± obtained CIs.

Statistical testing by t-tests. For correction of the multiplicity problem arising from multiple tests, the Holm–Šídák methods was used, all test results are given in Source Data file.

### ATPase assays

The ATPase activities of the protein samples were measured with a UV/Vis Spectrometer (Lambda 35, Perkin Elmer) using an ATP (2 mM, Jena Bioscience) regenerating system containing NADH (0.2 mM, Sigma-Aldrich), PEP (0.2 mM, Sigma-Aldrich) and PK/LDH (6 U/mL/10 U/ml, Carl Roth). The measurement was carried out at 37 °C and a wavelength of 340 nm. The slit was chosen to be 1 nm and measurement intervals were set to 1 second. As NADH shows strong absorption at 340 nm, its consumption (and in an indirect manner, ATP's consumption as well) can be monitored.

$1\,\mu M$ of $Hsp90_2$ (i.e. $2\,\mu M$ monomers) or $Cdc37_2$-$Hsp90_2$-fusion were used with $1\,\mu M$ or equimolar concentrations of the respective binding partners. After 20 min, radicicol (200 μM, Sigma) was added to inhibit Hsp90's ATPase activity. The inhibition curve was used as a reference and was subtracted from the data.

A one-way ANOVA was performed to compare the effect of nine different protein conditions on Hsp90's ATPase rate. The one-way ANOVA revealed that there was a statistically significant difference in mean ATPase between at least two conditions ($F(8,58) = [23.41464]$, $p = 0.05$). Tukey's HSD Test for multiple comparisons found the conditions whose mean ATPase rate were significantly different at a significance level of $p = 0.01$, $p = 0.05$ and $p = 0.1$, respectively. All conditions are shown in Supplementary Fig. 5, selected ones in Fig. 4.

### ATPase assays to check protein fusions' functionality

To check whether the introduction of the artificial linker alters the proteins' interactions, ATPase assays of $Hsp90_2$ only, $Cdc37_2$-$Hsp90_2$-fusion, and $Hsp90_2$ in the presence of freely added Cdc37 were compared both in the absence and presence of the client kinase Ste11 (Supplementary Fig. 5).

The addition of equimolar Cdc37 significantly reduces Hsp90's already low rate of $1.3 \pm 0.3$ to $0.7 \pm 0.3$ ATP/min. The linked $Cdc37_2$-$Hsp90_2$-fusion protein yields the very same result, clearly indicating that it is functional and that the protein interactions are not altered by the linker. The addition of equimolar Ste11 to Hsp90 also reduces its ATPase activity to $1.0 \pm 0.3$ ATP/min, although not significantly. However, when both Ste11 and Cdc37 are present, Cdc37's reducing effect on Hsp90's ATPase activity is abrogated. This holds true for the freely added Cdc37 as well as for the fusion-construct with rates of $1.1 \pm 0.4$ and $0.9 \pm 0.3$ ATP/min, respectively, indicating that Ste11 influences the interaction (and possibly also manner and/or position of binding) between Hsp90 and its cochaperone.

### Reporting summary

Further information on research design is available in the Nature Portfolio Reporting Summary linked to this article.

## Data availability

All recorded single-molecule and simulated traces are provided in Supplementary Information and Source Data files. Source data for all figures are provided as Source Data file. Source data are provided with this paper.

## Code availability

All software tools are available: SMACKS v1.4 at https://github.com/sciSonja/SMACKS, Hidden-Markury v1.0.1 at https://github.com/ChristianGebhardt/Hidden-Markury, MASH-FRET v.1.3.3.2 at https://github.com/RNA-FRETools/MASH-FRET. Additional code is available in the Supplementary Code file.

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

## Acknowledgements

This work was supported by the Deutsche Forschungsgemeinschaft (DFG) under Germany's Excellence Strategy (CIBSS EXC-2189 Project ID 390939984) (T.H.) and the SFB1381 programme (Project ID 403222702) (T.H.). We thank Aljaz Godec and Dmitrii Makarov for helpful discussions. We thank Marianne Birkle and Michael Witt for their support with protein purification.

## Author contributions

T.H. designed the research; J.S. and L.V. performed the measurements; B.H. made the protein constructs. J.S. and L.V. analysed the data after

consultation with T.H. All authors wrote the manuscript and have given approval to the final version of the manuscript.

## Funding

## Competing interests
The authors declare no competing interests.
