## [Peer Review File · Nature Communications]

Cochaperones convey the energy of ATP hydrolysis for directional action of Hsp90REVIEWER COMMENTS

Reviewer #1 (Remarks to the Author):

The molecular chaperon Heat shock protein-90 (Hsp90) is a dynamic protein and it occupies at least four states, two (using FRET efficiency) open states (0 and 1) and two closed states (2 and 3) which differ kinetically. The authors state that these four states are not coupled to the ATP hydrolysis (based on their previous published work).

In this paper the authors Vollmar et al have used single-molecule FRET and added several cochaperones (Aha1, Sba1, Cdc37) and the client kinase Ste11 to Hsp90. They then showed when the energy of ATP hydrolysis is coupled to Hsp90 conformational changes and consequently Hsp90 directionality. The authors suggest that the presence of Cdc37, Ste11 and ATP lead to changes of Hsp90 kinetic rates, but they do not introduce directionality. However, addition of Aha1 and Sba1 cause directionality. Therefore, all three cochaperones are necessary to convey the energy of ATP hydrolysis. This is an interesting story and provides evidence on how co-chaperones, a kinase client and ATP control Hsp90 dynamic in vitro. There are few issues that the authors need to address in order to strengthen their claims

Figure 2- The authors do not explain how data from single-molecule detection and analysis of dsDNA is applied and capable of quantifying directionality of Hsp90.

Surely, some Hsp90 mutants need to be tested in order to confirm these states.

Figure 3- Hsp90+Ste11+ATP control data is missing. In the cellular context Cdc37 is required for kinase binding to Hsp90. However, in vitro, these two proteins (Hsp90:kinase) are capable of interacting.

Figure 3- CDC37 phosphomimetic has been used in this experiment, however I am sure that the authors know very well that PP5 has to dephosphorylate these sites in order for the client to be released. Therefore, it is unclear how addition of Aha1 and P23 provides any relevance to understand the "directionality".

Figure 3d- The error bars for some of the presented data is very high. Has any statistical analysis been used to analyze this data?

Figure 4- Addition of Ste11 to Hsp90 and measuring the ATPase activity is missing. Previous work by Sophie Jackson's lab has shown that addition of client is sufficient to control Hsp90 ATPase activity.

Reviewer #2 (Remarks to the Author):

Observational directional cycling between conformational states in situations where there is no linear of angular movement is a difficult problem in biophysics. Here the authors attempt to observe directionality in conformational cyclin in Hsp90, in presence of various molecular cochaperones. The authors first test their analysis on an artificial smFRET system where directionality of transition between different FRET states is established by specific combinations of red and green lasers for specific time periods. Then, they use their pipeline to analyze smFRET data between FRET pairs placed on each monomer of Hsp90, in presence of various cochaperones. In presence of Cdc37, Ste11, and ATP, the authors observe no directionality. This is contrary to current models. However introduction of two additional cochaperones – Aha1 and Sba1 – lead to the overall DeltaG of the cycle to be ~ -2.1 kT, signifying directionality. Overall, this is a neat paper on directionality in conformational cyclin in an important ATPase. The findings may indeed be generally true of GHL NTPases.

The authors perform an artificial cycling of states by periodically cycling between different lasers for different time durations. Although they show in one case that their directional cycling (ground truth) matches what the model predicts, the authors really need to establish what the limit are on parameters such as the ratio of rate constants, time durations, etc, over which the model works. When would the model fail to predict directional cycling even though it is the ground truth? Conversely, how does the model perform when the ground truth is just a stochastic transition between 4 states with no directional bias?

On similar lines, I feel an important control to perform to verify if indeed the author's claim of directionality is true is to use an ATP analogue that cannot be hydrolyzed, or that hydrolyzes very slowly, as cyclical movement in a directional fashion between conformational states should only be possible in presence of ATP hydrolysis. This additional experiment would also tie in well to the author's discussion on how the conformational transitions may be linked to substeps in ATP hydrolysis.

The author's findings that directionality and ATPase rates are only weakly linked is quite interesting and would benefit for an additional discussion on weak coupling between structural transitions and chemical substeps in the cycles of GHK ATPases in general, as opposed to more well-studied P-loop NTPases. For example, see PMID 22484318.

Reviewer #3 (Remarks to the Author):

My review is uploaded as a 'review attachment'.

This paper delineates an interesting and quite original approach to identify directionality in protein functional cycles. The authors study the conformational transitions of Hsp90 in the presence of various co-chaperones at the single-molecule level. They use FRET signals to study whether detailed balance is maintained or a net flux is created in the set of transitions they observe. They conclude that the addition of three co-chaperones is necessary in order to push the ATP-driven cycle of Hsp90 out of equilibrium.

This work has the potential to become important and influential. However, in its current form it is not ready for publication and requires significant reworking, for the reasons delineated below.

1. The paper is written in a rather sloppy, perhaps hasty manner. Many experimental details are missing. These include details like the way data is treated, including aspects like the leak between channels, direct acceptor excitation, photobleaching etc. The authors use ALEX, but we are not shown how the acceptor excitation channel looks like, so we cannot tell whether, for example, a signal like the one in Figure 3c starting at 20 seconds is not due to a bleached acceptor. We are also not told how exactly the HMM analysis is done. For example, is it performed on each trace separately or on all traces together? Are the reported rates obtained directly from the HMM analysis, or from dwell-time analysis, like some people are doing? And how are parameter errors obtained from this analysis?
2. In general, very little data is shown in the paper. In fact, only two traces are shown. The authors do provide all the data in the form of txt files, but surely they do not expect the reader to plot all these files one by one. Please show a significant number of traces, e.g. in a supporting information file.
3. Missing from the paper is the very basic demonstration of the fact that a 4-state model is required to treat the data. We are sent to a 2016 paper, but that is not at all sufficient when this point is so central to the current paper. This issue should be directly shown in the paper and the reader should be convinced that indeed four states with two degeneracies are required and/or are optimal. What about other models? What happens if diagonal transitions (0-2 and 1-3) are included in the analysis?
4. The major control used by authors to show that they can detect directionality seems to be FLAWED. They conduct an experimental simulation of a directional cycle by alternating laser excitation on a FRET-labeled sample. However, they use a DETERMINISTIC set of times between switches of the excitation, and then analyze the data using HMM analysis. Yet in general, HMM analysis is derived for stochastic (kinetic) models, where there is a certain probability for switching between states and no deterministic switches are expected. I therefore do not see how their control data can be analyzed with HMM. The authors should re-perform the analysis using stochastic switching. Also, they should cover in their control the range of parameters that they get from the Hsp90 analysis, i.e. down to 2 K_BT, rather than just showing analysis with larger values (i.e. 3 and 10 K_BT).
5. The quantity shown in equation 2 is NOT entropy production. It is a quantity related to the thermodynamic force in a cycle (sometime also called Affinity), and indeed will be different than 0 when equilibrium is breached, but it should be referred to properly. See for example the book Free Energy Transduction and Biochemical Cycle Kinetics by Terrell

Hill, starting on page 12. The correct expression for entropy production in a cycle involving discrete states is:

$$\dot{S}(t) = \frac{1}{2} \sum_{i \neq j} (k_{ij}P_i - k_{ji}P_j) \ln \frac{k_{ij}P_i}{k_{ji}P_j}$$

Here the P s are of course the populations of the states.

The affinity CAN be used to study directionality in a cycle, but the correct name should be given. As a matter of fact, it would be interesting to perform not only a calculation of the affinity but also of the proper entropy production and observe that both of them are showing deviation from 0.

6. Panels b and c in Figure 3 show numbers of transitions (the legend refers to 'normalized amount'- what is this?)- it is not clear how these were calculated (from Viterbi?), but in any case they should be given for all data sets, and errors on these numbers should also be estimated (from repeats of the experiment). This is very important for analysis of the robustness of the results.
7. Additional minor issues:
 - a. What is actually shown in Fig. 2b? The sum of the two channels? A 'FRET signal'? If the latter, why is FRET efficiency not shown with the traces of Figure 3?
 - b. Please pay attention to the signs of energies, they are sometimes given as positive and sometimes as negative.
 - c. The first paragraph of the discussion is a bit difficult to understand. For example, it is not clear what the statement "This could for example be provided as binding free energy" means, or for that matter the term 'upstream equilibrium'. Please re-write in a more friendly manner.
 - d. Line 268- "our data clearly shows that there is no linear succession of binding events"- how is this inferred from the data?

REVIEWER COMMENTS

Replies in *blue*, changes and additions to the text of the manuscript in *brown*.
These changes are also highlighted in the main text **in green**.

Reviewer #1 (Remarks to the Author):

The molecular chaperon Heat shock protein-90 (Hsp90) is a dynamic protein and it occupies at least four states, two (using FRET efficiency) open states (0 and 1) and two closed states (2 and 3) which differ kinetically. The authors state that these four states are not coupled to the ATP hydrolysis (based on their previous published work).

In this paper the authors Vollmar et al have used single-molecule FRET and added several cochaperones (Aha1, Sba1, Cdc37) and the client kinase Ste11 to Hsp90. They then showed when the energy of ATP hydrolysis is coupled to Hsp90 conformational changes and consequently Hsp90 directionality. The authors suggest that the presence of Cdc37, Ste11 and ATP lead to changes of Hsp90 kinetic rates, but they do not introduce directionality. However, addition of Aha1 and Sba1 cause directionality. Therefore, all three cochaperones are necessary to convey the energy of ATP hydrolysis. This is an interesting story and provides evidence on how co-chaperones, a kinase client and ATP control Hsp90 dynamic in vitro. There are few issues that the authors need to address in order to strengthen their claims

Thank you for appreciating the importance of this study for co-chaperone, client and ATP control of Hsp90's dynamics. In response to your comments, we have strengthened our claims as follows:

Point 1: Figure 2- The authors do not explain how data from single-molecule detection and analysis of dsDNA is applied and capable of quantifying directionality of Hsp90. Surely, some Hsp90 mutants need to be tested in order to confirm these states.

Reply1: *We apologize that the reason for the dsDNA experiment was not clearly described: The data from dsDNA is a proof of concept that our method of retrieving directionality from single-molecule data works. Therefore, the underlying ground truth has to be known and this was achieved with a dsDNA sample that contained the same fluorophores as Hsp90. Fig. 2 shows that the ground truth (underlying laser trigger pattern) can well be retrieved from our single-molecule (fluorophore) data. We clarified this point in the revised manuscript by including the following sentences:*

“To address the challenge of observing directionality in conformational changes, we have developed a new testing procedure that uses external laser triggering to create artificial directionality in experimental data. Therefore, we created an artificial smFRET system using labelled lowFRET dsDNA (Hellenkamp et al. 2018). With a Total Internal Reflection Fluorescence (TIRF) microscope for data acquisition, and dwell time analysis and Hidden Markov Modelling for analysis and evaluation, we first checked whether directionality could be retrieved from single-molecule experiments. “

After having shown that directionality can be quantified from single-molecule experiments, we expanded our experiments to the HSp90 machinery. The dynamics of Hsp90 can be well described by a four state model. This has previously been shown in several publications, also for mutations: Mickler et al. 2009, <https://doi.org/10.1038/nsmb.1557>; Schmid et al. 2016,

<https://doi.org/10.1016/j.bpj.2016.08.023>; Schmid et al. 2020,
<https://doi.org/10.7554/eLife.57180>.

Additionally, we added Supplementary Fig. 3, which shows that we tested different models for our protein data. We summarize the results in the following new sentence in the manuscript:

“Hidden Markov Modelling (HMM) shows that Hsp90’s dynamics are best described by a four state model (see Supplementary Fig. 3.), having two open and two closed states.”

Altogether, the dsDNA control and the Hsp90 experiments (supported by simulations) demonstrate that we are capable of quantifying directionality.

Point2: Figure 3- Hsp90+Ste11+ATP control data is missing. In the cellular context Cdc37 is required for kinase binding to Hsp90. However, in vitro, these two proteins (Hsp90:kinase) are capable of interacting.

Reply2: *Thank you for pointing this out. We have now also measured the Hsp90+Ste11+ATP control and added this to the Fig. 3g. We find a small ΔG of -0.8 ± 0.3 $k_B T$, which further supports our conclusion that several cochaperones are needed to obtain a directional cycle.*

Point3: Figure 3- CDC37 phosphomimetic has been used in this experiment, however I am sure that the authors know very well that PP5 has to dephosphorylate these sites in order for the client to be released. Therefore, it is unclear how addition of Aha1 and P23 provides any relevance to understand the “directionality”.

Reply3: *Thank you for pointing out the importance of phosphorylation, which we have now investigated in more detail and discuss on a few points in the revised manuscript. We have done two experiments with Cdc37 without phosphomimetic (Cdc37_noMut). For the first experiment (Hsp90+Cdc37_noMut+Ste11+ATP) we see no directionality, as expected. More interesting was the second experiment (Hsp90+Cdc37_noMut+Ste11+ATP+Aha1+p23), where we did not have a clear expectation. Again we do not see a directionality, highlighting the importance of phosphorylation for this system. We have to work with phosphomimetics here because we work with recombinant yeast proteins produced in E. coli.*

Point4: Figure 3d- The error bars for some of the presented data is very high. Has any statistical analysis been used to analyze this data?

Reply4: *We have revised and extended our statistical analysis. We did two-sample t-tests and corrected for the multiplicity problem by the Holm-Šidák method. We now give selected p-values in Fig. 3g; p values and the testing procedure for all comparisons can be found in Supplementary Data file 3.*

Point5: Figure 4- Addition of Ste11 to Hsp90 and measuring the ATPase activity is missing. Previous work by Sophie Jackson’s lab has shown that addition of client is sufficient to control Hsp90 ATPase activity.

Reply5: *The ATPase rate of Hsp90 in presence of Ste11 is shown in Supplementary Fig. 6 in the 4th bar from the left. The addition of Ste11 does not change the ATPase rate of Hsp90 significantly (One-way ANOVA, $p = 0.1$). Therefore, Ste11 on its own does not control Hsp90’s ATPase activity. In the publication by Sophie Jackson’s lab (<https://doi.org/10.1006/jmbi.2001.5245>) this was shown for human Hsp90 with the GR-LBD as client. In comparison to the yeast homologue, the ATPase activity of human Hsp90 is even*

much lower. In this paper they also discuss, that partially unfolded clients (like the kinase Ste11 used by us) do not increase human Hsp90's ATPase activity. This already hints to different mechanisms for different types of clients. We added the following sentence to clarify this point:

“For human Hsp90, it was previously shown that the sole addition of a client (the ligand-binding domain of the glucocorticoid receptor) is sufficient to control Hsp90's ATPase activity (McLaughlin et al. 2002). However, the presence of equimolar amounts of Ste11 did not increase the ATPase of the Hsp90 yeast homolog used within this paper (Supplementary Fig. 6).”

Supplementary Fig. 6: Relative and absolute ATPase rates of yeast Hsp90 (2 μM) in presence and absence of cochaperones Cdc37, Aha1 and Sba1, and the client kinase Ste11. Hsp90₂ (1 μM, i.e. 2 μM monomers) is a slow ATPase with 1.3 hydrolysed ATP per minute (violet). The addition of free Cdc37 (2 μM, equimolar) further decreases the ATPase rate (dark blue, striped). As the Cdc37₂-Hsp90₂-fusion shows the same behaviour, the functionality of this construct (dark blue) is proven. Ste11 (2 μM) slightly decreases Hsp90's ATPase activity (green). Additionally, Ste11 hinders Cdc37's ATPase-decreasing effect on Hsp90 of both the freely added protein (middle blue, striped) as well as the Cdc37₂-Hsp90₂-fusion (middle blue). Aha1, Cdc37, Sba1 and Ste11 together (1 μM each, lightest blue) strongly increase the ATPase to 2.6 ATP/min. This effect is not achieved by only adding Cdc37 and Aha1 (1 μM each, darkest blue). Statistical significance tested by one-way ANOVA and Tukey post hoc test (described in methods).

Reviewer #2 (Remarks to the Author):

Observational directional cycling between conformational states in situations where there is no linear or angular movement is a difficult problem in biophysics. Here the authors attempt to observe directionality in conformational cycling in Hsp90, in presence of various molecular cochaperones. The authors first test their analysis on an artificial smFRET system where directionality of transition between different FRET states is established by specific combinations of red and green lasers for specific time periods. Then, they use their pipeline to analyze smFRET data between FRET pairs placed on each monomer of Hsp90, in presence of various cochaperones. In presence of Cdc37, Ste11, and ATP, the authors observe no directionality. This is contrary to current models. However introduction of two additional cochaperones – Aha1 and Sba1 – lead to the overall ΔG of the cycle to be ~ -2.1 kT, signifying directionality. Overall, this is a neat paper on directionality in conformational cycling in an important ATPase. The findings may indeed be generally true of GHL NTPases.

Thank you for appreciating this study and for pointing out a possible generality for GHL NTPases.

Point1: The authors perform an artificial cycling of states by periodically cycling between different lasers for different time durations. Although they show in one case that their directional cycling (ground truth) matches what the model predicts, the authors really need to establish what the limit are on parameters such as the ratio of rate constants, time durations, etc, over which the model works. When would the model fail to predict directional cycling even though it is the ground truth? Conversely, how does the model perform when the ground truth is just a stochastic transition between 4 states with no directional bias?

Reply1: *We have now simulated several more parameters close to what we measure for Hsp90 and also just stochastic transitions ($\Delta G=0$ k_BT). For the experiments stochastic switching is not yet possible, but the directional artificial cycling experiment shows that the succession of states can be detected from single fluorophores (see also our reply to point 4a of reviewer#3). Therefore, what remains to show is that ground truth can be recovered, which is well doable by simulations. We now have done five different simulations of single-molecule dynamics to cover more of the parameter space and to test if directionality can be retrieved in a wider parameter space. As we obtain a ΔG of around -2 k_BT from our Hsp90 experiments, we simulated data around this ΔG . In addition, the smallest rate for Hsp90 is 0.0022 Hz, therefore we created a parameter set with -2 k_BT and a smallest rate of 0.002 Hz. The ratio of the smallest to the biggest rate is then 1/335. Additionally, we also tested how the analyses behave, when we have equally distributed state populations – their size differing from the populations in the Hsp90-like condition. Finally, we also created a parameter set in detailed balance (stochastic transitions, $\Delta G = 0$ k_BT). Altogether, these experiments show that the relevant ΔG and rate constants can well be detected with our experiments and analysis. We have added Supplementary Fig. 1 and the following text to the manuscript:*

“To quantify the directionality that can be retrieved by our software SMACKS and to compare it to another software (Hidden-Markury (Gebhardt 2021)), we used simulated data with a directionality of 0 k_BT, -3 k_BT and -10 k_BT, respectively (Fig. 2e, f and Supplementary Fig. 1). Additionally, two data sets with the same ΔG value of -2 k_BT but with different transition rates (one with rates similar to rates measured for Hsp90, and the other with equally distributed state populations) were simulated (Fig. 2e, f). The data was simulated with MASH-FRET

(Börner et al. 2018) and then analysed with SMACKS and Hidden-Markury. Both programmes, which have previously been tested in a comparative study (Götz et al. 2022), were able to retrieve the pre-specified ΔG s. For $\Delta G = 0 \text{ k}_B\text{T}$, Hidden Markury obtained a ΔG of $0.505 \text{ k}_B\text{T}$, while SMACKS resulted in a ΔG of $0.02 \text{ k}_B\text{T}$. This important control confirmed that for data representing a system in detailed balance, none of the programmes overestimated the ΔG values of the ground truth, i.e. resulted in a falsely positive directionality. For $\Delta G = -2 \text{ k}_B\text{T}$ (Hsp90 like), Hidden Markury recovered 93 % of the ground truth while SMACKS recovered about 80 %. In all cases with a $\Delta G \neq 0$, the directionality was underestimated by a maximum of 25% . A certain underestimation is expected, as the result from Hidden Markov Modelling (HMM) is only a lower limit (Godec and Makarov 2023). Altogether, our findings show that we can distinguish systems in detailed balance from those with a directional flux – even if the analysed system does not show strong directionality, i.e. small ΔG s.”

Our data analysis is mainly limited by the recording rate of our experiment (i.e. 2 Hz).

Point2: On similar lines, I feel an important control to perform to verify if indeed the author’s claim of directionality is true is to use an ATP analogue that cannot be hydrolyzed, or that hydrolyzes very slowly, as cyclical movement in a directional fashion between conformational states should only be possible in presence of ATP hydrolysis. This additional experiment would also tie in well to the author’s discussion on how the conformational transitions may be linked to substeps in ATP hydrolysis.

Reply2: *Thank you for suggesting this additional experiment. We added the very slowly hydrolysable ATP analogue ATP γ S to Cdc37₁-Hsp90₂ + Ste11 + Aha1 + Sba1. As expected, ATP γ S induces more closing of Hsp90. A cyclical movement was not observable. In fact, a two state model with only one open and one closed state represented the data best. We show the results of the suggested experiment in the new Supplementary Fig. 5.*

Supplementary Fig. 5: An energy source (e.g. ATP hydrolysis) is necessary for directed movement through a conformational cycle. **A** FRET histogram of Cdc37₁-Hsp90₂ + Ste11 + Aha1 + Sba1 in presence of ATP (light blue, 998 traces) or the non-hydrolysable ATP analogue ATP γ S (dark blue, 435 traces). ATP γ S leads to more closing of Hsp90 ($83 \pm 2 \%$ closed) and few opening and closing events are observed. **b** Hidden Markov modelling in the presence of ATP γ S shows that these dynamics are best described by a two state model. The same models as in Supplementary Fig. 3 were tested. State 0 represents open Hsp90, state 1 closed Hsp90. **c** Transition rates for the two state model with 95 % confidence intervals. The opening rate (1 \rightarrow 0) is reduced compared to the opening rate in presence of ATP (2 \rightarrow 1, see Fig. 3e or Supplementary Fig. 4).

Point3: The author's findings that directionality and ATPase rates are only weakly linked is quite interesting and would benefit for an additional discussion on weak coupling between structural transitions and chemical substeps in the cycles of GHK ATPases in general, as opposed to more well-studied P-loop NTPases. For example, see PMID 22484318.

Reply3: Many thanks for pointing this out. The comparison to other GHKL ATPases is indeed very insightful and we added a sentence on GHKL ATPases in the introduction and the following paragraph to the discussion section:

"On a related note, it is worth comparing these findings to other members of the GHKL ATPases, especially DNA gyrase, which has a structurally highly similar ATP binding domain. Similarly to Hsp90's conformational behaviour, it was previously shown that gyrase's structural transitions are not strictly dependent on ATP hydrolysis but are rather loosely coupled (Bandhakavi et al. 2003). When processing DNA, both conformational states of the gyrase prevail in absence of the nucleotide. However, the presence of ATP accelerates the transition from one to the other and funnels the system towards a directional cycle through loosely coupled transitions (Basu et al. 2012). This fits nicely to our findings and our interpretation (Fig. 5), where weakly coupled transitions (equilibrium, left) are funnelled towards a directional cycle by Cdc37, Aha1, Sba1 and ATP. Such a loose coupling between structural transitions and chemical sub steps might therefore be a hallmark of GHKL ATPases in general, as opposed to the tightly coupled and well-studied P-loop NTPases (Guo et al. 2016)."

Reviewer #3 (Remarks to the Author):

This paper delineates an interesting and quite original approach to identify directionality in protein functional cycles. The authors study the conformational transitions of Hsp90 in the presence of various co-chaperones at the single-molecule level. They use FRET signals to study whether detailed balance is maintained or a net flux is created in the set of transitions they observe. They conclude that the addition of three co-chaperones is necessary in order to push the ATP-driven cycle of Hsp90 out of equilibrium. This work has the potential to become important and influential. However, in its current form it is not ready for publication and requires significant reworking, for the reasons delineated below.

Thank you for appreciating our work. We have performed several more experiments to strengthen our claims, as detailed below.

Point1. The paper is written in a rather sloppy, perhaps hasty manner. Many experimental details are missing. These include details like the way data is treated, including aspects like the leak between channels, direct acceptor excitation, photobleaching etc. The authors use ALEX, but we are not shown how the acceptor excitation channel looks like, so we cannot tell whether, for example, a signal like the one in Figure 3c starting at 20 seconds is not due to a bleached acceptor. We are also not told how exactly the HMM analysis is done. For example, is it performed on each trace separately or on all traces together? Are the reported rates obtained directly from the HMM analysis, or from dwell-time analysis, like some people are doing? And how are parameter errors obtained from this analysis?

Reply1: Our apologies that we have built too much on our previous publications with respect to data analysis. We now show the acceptor signal upon acceptor excitation in Figure 3. The correction of the single-molecule traces is now described in more detail in the methods section. We corrected our data accordingly, but please note, that this would not be necessary here (that is why we did not focus on that), because we only analyse conformational dynamics and not absolute FRET efficiency values or distances here. Both HMM software (SMACKS and Hidden-Markury) applied use directly the donor and acceptor fluorescence upon donor excitation and not the FRET efficiency as an input. We have now significantly extended the supplementary material and provided a full description of the data analysis by both HMM software in the supplementary methods.

Point2. In general, very little data is shown in the paper. In fact, only two traces are shown. The authors do provide all the data in the form of txt files, but surely they do not expect the reader to plot all these files one by one. Please show a significant number of traces, e.g. in a supporting information file.

Reply2: We show now several more traces in the supplementary material in Supplementary Fig. 2, which is shown here:

Supplementary Fig. 2: Exemplary single-molecule FRET traces showing Hsp90's opening and closing dynamics in presence of various cochaperones. Viterbi path (black) obtained for the part of the trace before photo bleaching (faded colors indicate at least one dye being bleached). FRET signal (acceptor emission after donor excitation) shown as red line, direct excitation of the donor shown in green, and of the acceptor in grey. **a-f** Cdc37₁-Hsp90₂ + ATP + Ste11 + Aha1 + Sba1. **g-l** Cdc37₁-Hsp90₂ + ATP + Ste11.

Point3. Missing from the paper is the very basic demonstration of the fact that a 4-state model is required to treat the data. We are sent to a 2016 paper, but that is not at all sufficient when this point is so central to the current paper. This issue should be directly shown in the paper and the reader should be convinced that indeed four states with two

degeneracies are required and/or are optimal. What about other models? What happens if diagonal transitions (0-2 and 1-3) are included in the analysis?

Reply3: We now show in Supplementary Fig. 3 (see below) how several models of different complexity perform. All these models were tested for all experimental conditions. A 4 state model with two degeneracies fits the single-molecule traces indeed the best and was chosen based on the Bayesian information criterium (BIC, Fig. S3h). When diagonal transitions (0-2 and 1-3) are included they are always close to zero (Supplementary Fig 3k and l).

Supplementary Fig. 3: Selecting the best model describing Hsp90's conformational behaviour. The ensemble Hidden Markov modelling for all traces of one experiment (here: Cdc37₁-Hsp90₂ + ATP + Ste11 + Aha1 + Sba1, 291 traces) was carried out with models (a-g) of varying complexity and amounts of hidden states. High FRET states in grey. **h** A four state model best fits the data, having the lowest Bayesian information criterion (BIC). Model a and b have an identical BIC of 777690. **k, l** Representations of model a and b with circle sizes proportional to the state populations and arrow widths proportional to the transition rates. **m** Transition rates with 95 % confidence intervals of model k and l. Diagonal rates are close to zero. Calculations were done with SMACKS as described in the methods and Supplementary methods.

Point4a. The major control used by authors to show that they can detect directionality seems to be FLAWED. They conduct an experimental simulation of a directional cycle by alternating laser excitation on a FRET-labeled sample. However, they use a DETERMINISTIC set of times between switches of the excitation, and then analyze the data using HMM analysis. Yet in general, HMM analysis is derived for stochastic (kinetic) models, where there is a certain probability for switching between states and no deterministic switches are expected. I therefore do not see how their control data can be analyzed with HMM. The authors should re-perform the analysis using stochastic switching.

Reply4a: *Thank you for this helpful comment, which led to several very interesting discussions with theoreticians during the last weeks. We agree that for deterministic signals no HMM analysis should be done, as HMM is derived for stochastic models. But HMM analysis is successfully applied to non-stochastic signals, e.g. in speech recognition. As far as we can see, it is unclear why it works so well in speech recognition and also for our deterministic single-molecule fluorescence trace. We and others believe that this is a very interesting finding and we therefore like to keep the HMM analysis of the deterministic trace as a supplementary note (see below).*

In the main text we have now done an analysis, which does not rely on stochastic models. We now use a threshold to separate the two visible states and then do a dwell time analysis, which nicely reproduces the input rates (Fig. 2c,d).

We agree that the best way would be to have stochastic laser triggering (switching), but this currently is not possible with our setup for technical reasons and would need a complete redesign of the electronics and software for triggering. Fortunately, the purpose of this control was not to show that HMM analysis on single-molecule FRET trajectories works (which has been shown many times before), but that we can extract a directed succession of states, even if some of them are hidden, from our single-molecule trajectories. We are convinced that our control, including the new analysis, well serves this objective.

We have now added the new analysis to Fig. 2c,d and a new supplementary Note 1 with the HMM analysis including an explanation of all the caveats.

Point4b. Also, they should cover in their control the range of parameters that they get from the Hsp90 analysis, i.e. down to 2 K_BT, rather than just showing analysis with larger values (i.e. 3 and 10 K_BT).

Reply 4b: *As the reviewer suggested we have extended the parameter sets used in our simulations. In particular, we created two data sets with $\Delta G = -2$ k_BT. The first data set is based on the rate constant values and state populations we obtained from Hsp90 (smallest rate = 0.002 Hz). To test for robustness, we introduced a second data set with -2 k_BT with equally distributed state populations, which significantly differs from the populations in the Hsp90-like condition. For both cases, the analyses were able to retrieve the given directionality (Fig. 2f). Additionally, we added a condition without directionality, i.e. which is in detailed balance ($\Delta G = 0$ k_BT). In the Supplementary Fig. 1 we show all five conditions. You might notice, that the analyses now also perform better for the -10 k_BT and -3 k_BT case. This is not caused by an improvement in analysis, but is due to a parameter import error during trace simulation with MASH-FRET. This bug was now fixed due to our issue report. You can follow this exchange on GitHub (<https://github.com/RNA-FRETools/MASH-FRET/issues/104>).*

Point5. The quantity shown in equation 2 is NOT entropy production. It is a quantity related to the thermodynamic force in a cycle (sometime also called Affinity), and indeed will be different than 0 when equilibrium is breached, but it should be referred to properly. See for example the book Free Energy Transduction and Biochemical Cycle Kinetics by Terrell Hill, starting on page 12. The correct expression for entropy production in a cycle involving discrete states is:

$$\dot{S}(t) = \frac{1}{2} \sum_{i \neq j} (k_{ij}P_i - k_{ji}P_j) \ln \frac{k_{ij}P_i}{k_{ji}P_j}$$

Here the Ps are of course the populations of the states. The affinity CAN be used to study directionality in a cycle, but the correct name should be given. As a matter of fact, it would be interesting to perform not only a calculation of the affinity but also of the proper entropy production and observe that both of them are showing deviation from 0.

Reply5: *Thank you for your correction and the detailed description, that was very helpful. You are completely right, that the terms entropy production and thermodynamic force (ΔG) and affinity cannot be used interchangeably. We corrected this now and report both values: ΔG and the entropy production. These values and the flux and the errors are now shown for all conditions in Supplementary Tab. 2.*

Point6. Panels b and c in Figure 3 show numbers of transitions (the legend refers to 'normalized amount' - what is this?) - it is not clear how these were calculated (from Viterbi?), but in any case they should be given for all data sets, and errors on these numbers should also be estimated (from repeats of the experiment). This is very important for analysis of the robustness of the results.

Reply6: *We are sorry that we did not explain the analysis in detail. We have now detailed this in the supplementary methods. In short: The number of transitions was calculated based on the transition matrices obtained from HMM analysis. The transitions can be calculated by multiplying the stationary state populations vector with the transition matrix. The diagonal of the resulting matrix was set to zero as staying in one's initial state does not result in an observable transition. The amount of transitions was then normalized to 100. Errors for the transitions were calculated from repeats of the experiments. Number of transitions, of repeats and errors for all protein conditions are now given in Supplementary Tab. 2.*

7. Additional minor issues:

Point7a. What is actually shown in Fig. 2b? The sum of the two channels? A 'FRET signal'? If the latter, why is FRET efficiency not shown with the traces of Figure 3?

Reply7a: *The signal shown in Fig. 2b is the acceptor emission after excitation (excited with either only the donor laser or both lasers, according to the sequence shown in Fig. 2a to obtain artificial high FRET). We clarified the label in Fig. 2b.*

The FRET Efficiency is not shown in Fig. 3a and c, because it is not used for analysis. A big advantage of SMACKS (and also Hidden-Markury) is, that a 2D analysis is performed. Not only 1 signal is used (the FRET E), but two anticorrelated signals (donor emission after donor excitation and acceptor emission after donor excitation).

Point7b. Please pay attention to the signs of energies, they are sometimes given as positive and sometimes as negative.

Reply7b: *Thanks for pointing this out. We have now double-checked the signs of energies to be consistent.*

Point7c. The first paragraph of the discussion is a bit difficult to understand. For example, it is not clear what the statement “This could for example be provided as binding free energy” means, or for that matter the term ‘upstream equilibrium’. Please re-write in a more friendly manner.

Reply7c: *We rewrote the first paragraph of the discussion, which reads now:*

“Most suggested conformational cycles of Hsp90 imply strong directionality as they are commonly depicted as being unidirectional. Here we show a fundamentally different scenario, in which an upstream (i.e. preceding) equilibrium allows for an assembly of most of the involved proteins, and then one (or few) directed steps stabilize the functional complex (Fig. 5). We consider this scenario to be much more likely for the formation of any multi-protein complex for several reasons. First, for an energetic reason: Consider a complex of five molecules with a defined binding order. To achieve a reasonably defined binding sequence, the energy equivalent of at least one ATP hydrolysis (around 20 $k_B T$) needs to be consumed in each step, i.e. the hydrolysis of 5 ATP in total. Another possibility would be to provide this energy in the form of free energy of binding. However, this would eventually result in a very stable multiprotein complex that could only be broken apart again upon an energy input corresponding to five ATPs – one for every step – or by degradation of some of the proteins involved. In our opinion, both options seem unlikely, as they would be highly inefficient and a Hsp90 dimer binds (and hydrolyses) two ATP at maximum per cycle. In contrast, an upstream equilibrium preceding one or few directed steps, as suggested here, would only involve energies of a few $k_B T$ s. The final formation of the functional complex would then involve only stabilizing energy of e.g. one ATP hydrolysis. Therefore, the components could be recycled by the hydrolysis of one or two ATP, which are available from Hsp90.”

Point7d. Line 268- “our data clearly shows that there is no linear succession of binding events”- how is this inferred from the data?

Reply7d: *We clarified this by adding the following sentences:*

“A linear succession of binding events (i.e. breaking of detailed balance) is only possible in two ways: a) In a cyclic manner, with the energy source being coupled to the cycle, which was not the case here as it would need more energy than the observed $-2.1 k_B T$, and b) by the production and degradation of a protein for every step in the linear succession. This would be an extreme waste of energy and therefore considered highly unlikely.”

REVIEWER COMMENTS

Reviewer #1 (Remarks to the Author):

The authors have done an outstanding job conducting additional experiments and providing extra data to address mine and the reviewers' comments and concerns.

Although there are no issues with the quality and the importance of the data, however I strongly believe that the authors are unintentionally misinterpreting their data.

This is evident in Figures 3e, f g and Supplementary Fig. 4. These data suggest that ATP alone can provide directionality to Hsp90 whereas this does not occur with addition of Cdc37. The data also suggest that the magnitude of the directionality is increased with the additional all co-chaperones and the client Ste11.

Reviewer #2 (Remarks to the Author):

I very much appreciate the way in which the authors have addressed my comments. The authors added simulations strengthen their claim that they can distinguish detailed balance from directional flux. The inclusion of the experiment in presence of ATP gamma S is great because it now to authors indeed show that in absence of regular ATP hydrolysis, no directional flux is observed. Finally, I appreciate the authors taking my suggestion and including a discussion on the generality with other GH1 NTPases.

Reviewer #3 (Remarks to the Author):

The authors have made much effort to address my comments, and I appreciate that. I think the paper is now better and would like to recommend publication, though I still have a few points to raise with the authors:

1. BIC analysis and the four-state model: the authors seem to give only relative BIC values in Supp. Fig. 3. It is important to see the absolute values, as the absolute differences between models might be quite small. How many trajectories were used to calculate these BIC values? As I look at the two four-state models, they seem very similar, certainly in terms of their relative BIC value. The authors dismiss the fully connected model base on small rates. However, just judging from the arrows in supp. Fig. 3, the rates for the transition 0 to 3 and back are also small. Therefore, it seems to me that at the very least the authors should repeat all calculations in the paper with the connected model (shouldn't be too much work) and show that the deviation from detailed balance they seem to detect is also maintained in this case.

2. The final model the authors present in Figure 5 is based on the idea that the binding of Aha1 and Sba1 is coupled to ATP hydrolysis, which drives the whole system out of equilibrium. But what is the evidence that this is the case? In fact, if I understand correctly, Hsp90 hydrolyzes ATP even in the absence of these two proteins.

3. I disagree that speech is deterministic. Rather, in speech recognition one encounters multiple options that can be assigned probabilities and then analyzed with statistical tools like HMM. In fact, I couldn't find in the two references provided by the authors any declaration that speech is non-stochastic. On the other hand, the signals that the authors generate for their control ARE fully deterministic, and therefore their whole analysis of the control signals seems to be imperfect. While I would not see this as a fatal flaw for the paper, I think that, at the very least, the authors should remove the declaration that speech recognition is not stochastic.

Point-by-point response to the reviewers' comments

(Reviewer comments in black, response in blue)

Reviewer #1 (Remarks to the Author):

The authors have done an outstanding job conducting additional experiments and providing extra data to address mine and the reviewers' comments and concerns.

Although there are no issues with the quality and the importance of the data, however I strongly believe that the authors are unintentionally misinterpreting their data.

This is evident in Figures 3e, f g and Supplementary Fig. 4. These data suggest that ATP alone can provide directionality to Hsp90 whereas this does not occur with addition of Cdc37. The data also suggest that the magnitude of the directionality is increased with the additional all co-chaperones and the client Ste11.

Response #1: Thank you for this comment, which shows that we still were not clear in every point of our data interpretation. First, and most importantly, we did not misinterpret the data. Second, we now discuss the lower limit for directionality in detail. This lower limit comes from four experiments, where we did not add any energy source to the system, i.e. where we have detailed balance, i.e. which means that $\Delta G = 0$ by definition. As in these experiments we measured ΔG s ranging from 0.3 kT to -0.94 kT, we only consider any experiment with a $|\Delta G|$ above 1 kT as clearly directional. Please note that the type of sign (positive or negative) only indicates the direction of the cyclic movement (clockwise or counter-clockwise), but has nothing to do with the question if a cycle is directional or not. This means that, for example, a value of -2 kT is just as directional as a value of +2 kT – only the sequence of states is reversed. Therefore, altogether, Hsp90+ATP is as little directional as Hsp90+ATP+Cdc37. Your last sentence is perfectly correct and the main message of our manuscript.

We have clarified these points by adding the following sentence to the manuscript “In experiments without energy source present, i.e. conditions where no directionality is possible, we measured ΔG values ranging from 0.3 to -0.9 k_BT (see Supplementary Tab. 2). Therefore, these measurements serve as the lower limits of what can be considered directional, i.e. we consider any experiment with a $|\Delta G|$ above 1 k_BT as clearly directional. Please note that the type of sign (positive or negative) indicates the direction of the cyclic movement (clockwise or counter-clockwise).” and by adding the following table with all former values of measurements without energy source to the supplement.

	Hsp90 ₂	Hsp90 ₂ + Ste11	Cdc37 ₁ -Hsp90 ₂	Cdc37 ₁ -Hsp90 ₂ + Ste11
ΔG [k _B T]	-0.9	0.27	-0.4	0.34
95% CI ΔG [k _B T]	[-0.5;1.5]	[0.25;0.28]	[-1.0;0.7]	[0.08; 0.88]
N _{sm-traces}	144	108	206	354

Reviewer #2 (Remarks to the Author):

I very much appreciate the way in which the authors have addressed my comments. The authors added simulations strengthen their claim that they can distinguish detailed balance from directional flux. The inclusion of the experiment in presence of ATP gamma S is great because it now to authors indeed show that in absence of regular ATP hydrolysis, no directional flux is observed. Finally, I appreciate the authors taking my suggestion and including a discussion on the generality with other GHL NTPases.

Response #2: Thank you 😊

Reviewer #3 (Remarks to the Author):

The authors have made much effort to address my comments, and I appreciate that. I think the paper is now better and would like to recommend publication, though I still have a few points to raise with the authors:

1. BIC analysis and the four-state model: the authors seem to give only relative BIC values in Supp. Fig. 3. It is important to see the absolute values, as the absolute differences between models might be quite small. How many trajectories were used to calculate these BIC values? As I look at the two four-state models, they seem very similar, certainly in terms of their relative BIC value. The authors dismiss the fully connected model base on small rates. However, just judging from the arrows in supp. Fig. 3, the rates for the transition 0 to 3 and back are also small. Therefore, it seems to me that at the very least the authors should repeat all calculations in the paper with the connected model (shouldn't be too much work) and show that the deviation from detailed balance they seem to detect is also maintained in this case.
2. The final model the authors present in Figure 5 is based on the idea that the binding of Aha1 and Sba1 is coupled to ATP hydrolysis, which drives the whole system out of equilibrium. But what is the evidence that this is the case? In fact, if I understand correctly, Hsp90 hydrolyzes ATP even in the absence of these two proteins.
3. I disagree that speech is deterministic. Rather, in speech recognition one encounters multiple options that can be assigned probabilities and then analyzed with statistical tools like HMM. In fact, I couldn't find in the two references provided by the authors any declaration that speech is non-stochastic. On the other hand, the signals that the authors generate for their control ARE fully deterministic, and therefore their whole analysis of the control signals seems to be imperfect. While I would not see this as a fatal flaw for the paper, I think that, at the very least, the authors should remove the declaration that speech recognition is not stochastic.

Response #3: Thank you for acknowledging our efforts, your comments were very helpful for further interesting discussions.

1. In the Supplementary Fig. 3 we have given the absolute differences in BIC. The BIC of the 4-states model is given in the figure caption (777690), as well as the number of trajectories used (291 traces). The reason we plotted the differences is to make small changes in the models' BICs clearly visible.
We repeated all calculations with fully connected models and always obtained the same BIC values as for the respective model with nulled diagonals. The obtained ΔG s differ only slightly (see the new Supplementary Fig. 3n, shaded bars). You can see that the overall finding of our publication is maintained and even supported.
2. Yes, this is one main message of our study (possibly surprising): Although Hsp90 also hydrolyses ATP in absence of cochaperones, this ATP hydrolysis is only coupled to Hsp90's N-terminal movements in the presence of those cochaperones. The evidence for the given model is that we only see directionality when Sba1 and Aha1 are present.
3. Thanks again for having started the discussion about determinism in time series. This is a very interesting topic and we will follow up on this. We removed the sentence stating that speech recognition is not stochastic. We nevertheless believe that there is a strong analogy between our problem and the speech recognition problem and – in the future – we will look closer into that. We discussed this issue again with a renowned theoretician, who gave the following example: "If you transmit a message "aaaaabbaaaaabb", it can be thought of as fully

deterministic – just like our deterministic sequence of pulses – it’s just a deliberately designed sequence of bits. When spoken aloud, the speech recognition software on a smartphone will decode it – it’s true that the person speaking is adding “noise” to the message, but that’s completely analogous to the noise produced by the stochasticity of photons in our experiments. And conversely, no stochastic HMM with whatever probabilities has - to our knowledge - ever generated a meaningful paragraph.” This stimulating discussion shows that this is a relevant topic worth investigating in future studies.